# Utilizing Real-Time Heart Rate Variability during Psychological Intervention Program for Complex Post-Traumatic Stress Disorder: A Case Study †

Bohye Im [1], Jooyoung Keum [2], Taeeun Kim [2], Kyo-il Lee [3,*] and Kyo-in Koo [2,4,*]

1   Daegu Maum-in Counseling Centre, Daegu 42003, Republic of Korea; peach2141@hanmail.net
2   Department of Electrical, Electronic and Computer Engineering, University of Ulsan,
    Ulsan 44610, Republic of Korea
3   Department of Community Development & Welfare, University of Daegu,
    Gyeongsan-si 38453, Republic of Korea
4   Basic-Clinical Convergence Research Institute, University of Ulsan, Ulsan 44610, Republic of Korea
*   Correspondence: rewaev@naver.com (K.-i.L.); kikoo@ulsan.ac.kr (K.-i.K.);
    Tel.: +82-53-850-4584 (K.-i.L.); +82-52-259-1408 (K.-i.K.)
†   The authors proposed and researched a new intervention program for complex post-traumatic stress disorder
    as well as how to use real-time heart rate variability during psychological intervention programs. This article
    focuses on describing HRV use for researchers in applied sciences. The authors believe that the psychological
    aspect is worth introducing to Korean psychological society. The psychological part has been published in the
    *Korean Journal of Counseling: Case Studies and Practice* 2022, titled "A pilot study to verify the effect of the
    intervention of the complex post-traumatic stress disorder symptoms relief program based on phase-based
    approach: A single case study" in the Korean language (doi: 10.15703/kjccsp.7.2.202212.55).

**Abstract:** This study explores the use of real-time heart rate variability (HRV) monitoring as an innovative tool in a psychological intervention program for complex post-traumatic stress disorder (CPTSD). The research focuses on a 25-year-old male subject, presenting severe symptoms of CPTSD resulting from prolonged exposure to traumatic events. The intervention program, conducted over four sessions, integrated cognitive and physical therapies, including bottom-up and top-down approaches. Real-time HRV data, reflecting autonomic nervous system activity, were collected using a wearable heart rate sensor and analyzed alongside qualitative data from session transcripts and observational notes. The findings demonstrated a significant correlation between HRV changes and the subject's psychological state during therapeutic interventions. Key HRV metrics, such as the change ratio of the RMSSD and SDNN, responded notably to traumatic event recounting and stabilization techniques, suggesting their potential as indicators of emotional and physiological states during therapy. The study, while limited by its single-subject design, lays the groundwork for further investigations into HRV-assisted psychotherapy for a broader CPTSD patient cohort.

**Keywords:** post-traumatic stress disorder; complex post-traumatic stress disorder; real time; heart rate variability; psychological intervention program

## 1. Introduction

Experiencing a situation in which one's life is threatened, or witnessing a scenario in which others' lives are in danger, are considered traumatic events. Some traumatic events are one-time or short-term events, such as traffic accidents or assaults. Other traumatic events may occur repeatedly over a long period, such as child abuse and domestic violence. A one-time or simple traumatic event is called simple trauma, while recurring and persistent traumatic is referred to as long-term trauma [1–3].

Individuals who experience long-term trauma are more impulsive and have more difficulty in emotional regulation than those who experience simple trauma [1,4]. Repressed hostility may appear as somatization symptoms or, in extreme cases, may be accompanied

by dissociative identity disorder. Chronic complex trauma influences the severity of symptoms and the personality structure. This is not clearly explained by post-traumatic stress disorder (PTSD). In other words, symptoms due to complex trauma are characterized by the individual experiencing difficulties in self-organization (DSO), such as problems with emotional regulation, having a negative self-concept, and difficulties with interpersonal relationships, along with re-experiencing the trauma and demonstrating avoidance and hyper-arousal, which are major symptoms of PTSD. This syndrome is called complex post-traumatic stress disorder (CPTSD) [1,5].

Previous studies have involved separately diagnosing complex PTSD to elucidate the psychopathology and enable more effective treatment of subjects who have experienced repeated and chronically extreme traumatic events [1,4,6–10]. CPTSD is listed separately from PTSD in the International Classification of Diseases 11th Revision (ICD-11); however, it is not separately listed in the Diagnostic and Statistical Manual of Mental Disorders 5th Revision (DSM-5). Studies comparing CPTSD and PTSD have been undertaken and have produced evidence for intervention methods through worldwide clinical trials.

In the Republic of Korea, the Daegu subway disaster in 2003 drew attention to PTSD, and the National Trauma Center was established in 2018 after the Sewol ferry disaster in 2014. However, there was no discrimination between PTSD interventions and CPTSD interventions. Ahn introduced CPTSD in 2007 [11], emphasizing the requirement for intervention specific to CPTSD only. Lee reported psychotherapy and treatment for CPTSD in 2020 [6]. Studies on CPTSD interventions have increased in Korea recently. These recent studies have some limitations in that existing psychological treatments were applied to CPTSD and treatment based on CPTSD symptoms was not performed [12].

CPTSD requires psychological stabilization through physical stabilization and the setting of treatment goals for cognitive intervention. The condition induces emotional changes through cognitive distortion owing to the subject's experience of long-term trauma [8,13]. Because these emotional changes cause the subject to re-experience negative feelings and to dissociate symptoms from sensing, psychological stabilization by means of physical stabilization is required. CPTSD caused by continuous exposure to abuse and violence in interpersonal relationships negatively affects the subject's daily life through generating negative concepts about themselves and others and causing a distorted understanding in relational contexts. To review and restructure damaged interpersonal relationships, physical intervention as well as cognitive intervention should be applied for effective treatment.

To modulate the abnormal coping mechanisms of the brain caused by CPTSD, the intervention program designed in this study applies the principles of bottom-up intervention, top-down intervention, and integration intervention. The bottom-up approach is intended to change the lower part of the brain through experience from five sensors so that it eventually affects the upper part of the brain, such as the prefrontal cortex (PFC), which is responsible for cognition. The top-down approach is intended to change the bodily and emotional responses (the lower part of the brain) through reconstructing cognition (the upper part of the brain). In other words, the top-down approach is applied to the cerebral neocortex. It addresses cognition by seeking meaning and understanding to facilitate control of the affective disorder and bodily sensory experiences and to increase linguistic self-esteem [13]. The proposed intervention program consists of both the bottom-up approach and the top-down approach to treat CPTSD symptoms, which helps to stabilize the arousal state of the autonomic nervous system (ANS) and to restructure negative cognition and thinking.

In psychology, the effectiveness of an intervention program is typically evaluated based on the subject's subjective response to scaled questionnaires. Although clinical scales have been widely utilized for their validity and reliability, there are some limitations in consistency with this approach because it is a subjective self-report method. In biomedical engineering and digital healthcare fields, heart rate variability (HRV), which is a quantitative measure of the variation in intervals between heartbeats, has been widely utilized to quantify the state of the subject [14,15]. In psychology, it has been increasingly employed

to assess the effects of psychotherapy [16,17]. However, HRV has rarely been utilized for CPTSD.

HRV serves as a sensitive indicator of the functionality of the autonomic nervous system (ANS) [18–20]. This system's parasympathetic and sympathetic branches modulate the heart rate via influences on the sinoatrial node pacemaker [21]. The high-frequency (HF) peak of HRV, derived from spectral analysis of inter-beat intervals, is believed to represent the parasympathetic or vagal tone, though there is some debate regarding the sensitivity and specificity of commonly utilized HRV measures [22,23]. Reduced HRV, which is often observed in post-traumatic stress disorder (PTSD), implies autonomic inflexibility, potentially resulting from sympathetic overactivity, parasympathetic deficiency [20,21,24,25], or the exacerbation of common cardiovascular issues associated with PTSD [26]. Lower HRV values have also been noted in various psychiatric disorders, including schizophrenia, depression, bipolar disorder, and panic disorder [27–30], linking decreased HRV to pathophysiology, psychopathology, and increased mortality [21,31].

Therefore, this study investigated the relationship between HRV and the psychological status of one subject who suffers from CPTSD. The principal hypothesis was that checking the subject's HRV in real time could help a psychotherapist to determine the subject's psychological status.

## 2. Case Description and Methods

The protocol for this study was approved by the Institutional Review Board of the University of Ulsan. All procedures were performed in accordance with the relevant guidelines and regulations. The trial conformed to the tenets of the Declaration of Helsinki. The subject provided written informed consent for the publication of their case details.

### 2.1. Case Description, Diagnosis, and Etiology

The subject of this study was a 25-year-old male. His body mass index was in the overweight range. He had been a nonsmoker and had experienced no cardiac disease. He graduated from high school and was unmarried. Although he had close friends, there were some conflicts with them. His current level of physical health and overall life satisfaction were low. Prior to participating in this program, he had engaged in twelve psychotherapy sessions with the topics of family and interpersonal conflict and trauma based on CBT and the object relations psychology model. He stated that previous psychotherapy was helpful in enhancing his self-understanding and interpersonal coping. He complained about difficulties in interpersonal relationships, nightmares, and negative emotions due to school and family violence he had repeatedly experienced since childhood. However, when participating in this study, the subject self-reported re-experiencing avoidance, sensing threat, affect dysregulation, having a negative self-concept, relationship disturbances, etc. These symptoms are well-known as major features of CPTSD [32]. In Korean societies, diagnostic criteria for discriminating CPTSD symptoms from PTSD have not yet been developed [12]. To ensure the CPTSD representativeness of the participant, the authors requested an expert with a doctorate degree in educational psychology and a counseling expert certificate from the Korean Counseling Association (KCA) to compare and analyze the CPTSD diagnosis criteria. After a detailed evaluation based on the MMPI-2, interview data, voice record, and transcripts, the expert confirmed that the participant would be a representative case of CPTSD.

### 2.2. HRV Monitoring during the Intervention Program

We monitored the subject's HRV in real-time using a wearable heart rate sensor (H10, Polar, Finland) and a communication application (Elite HRV, Elite HRV Inc., USA) during the intervention program. The wearable sensor used in the investigation has been validated as a replacement for a Holter monitor in many previous reports [33–37]. After the intervention program, using the recorded inter-beat intervals, the statistical relation

between the interventional events during the program and the change ratio in the HRV was analyzed to test our hypothesis.

### 2.3. Proposed Intervention Program for CPTSD

The proposed intervention program for CPTSD in this study was designed using the imagery and cognitive interventions of cognitive behavioral therapy (CBT) [12] and stabilization treatments for body-based sensory processes. In addition, a phase-based approach, which is the recommendation of the International Society for Traumatic Stress Studies (ISTSS) for CPTSD treatment [38], was utilized. Each stage was structured as follows: Phase 1 was bottom-up stabilization training; Phase 2 was trauma treatment, and cognitive (bottom-up), somatosensory and imagery (top-down) techniques were applied; Phase 3 was the process of integration. Our program consisted of four sessions of 90 min each. The proposed program consisted of the consideration of complex PTSD core traumatic events and symptom differentiation, cognitive restructuring and self-management training, and use of a stabilization technique.

Session 1 introduced the program, assessed the symptoms and the trauma events of the participant, and implemented stabilization techniques. Session 2 examined CPTSD education and cognitive distortion correction. Session 3 covered how to use the sense of preference to improve adaptation to daily life. Session 4 discussed ways to check and maintain changes. Bottom-up stabilization training was conducted every session so that the participant could acquire self-stabilization skills. When invasion occurred due to traumatic memory processing, the first stage stabilization technique was implemented so that the treatment stages could interact organically. This is in accordance with the recommendations of the ISTSS [38].

To increase the content validity and effectiveness of this program, it was cross-checked by a mental health social worker supervisor, a disaster mental health specialist, and an eye movement desensitization and reprocessing movement specialist. The researcher who conducted this program is a psychotherapist with 14 years of experience in neuro-linguistic programming based on CBT, 4 years of firefighter psychotherapy experience, and 13 years of experience as a couple and family therapy specialist.

A simple time-series design method was used for this investigation to evaluate the effect of the proposed intervention program for CPTSD. A total of four sessions were conducted from 5 November 2020 to 25 November 2020. A preliminary survey was performed on 5 November 2020 before the main intervention program. An end-of-program survey was conducted on 25 November 2020 immediately after the last session. A follow-up survey was performed on 23 September 2021, almost ten months after the last session. During each session, the subject wore a heart rate sensor with a communication application to measure their HRV.

### 2.4. Data Analysis

2.4.1. Quantitative Analysis

The preliminary survey, end-of-program survey, and follow-up survey were conducted using a questionnaire. The questionnaire comprised the Impact of Event Scale–Revised (IES-R-K), which was first developed by Weiss and Marmar in 1997 [39] and was enhanced by Eun et al. in 2005 [40]. It additionally included the Athens Insomnia Scale (AIS) developed by Soldatos et al. in 2000 [41]. The IES-R-K is a self-report scale for trauma-related symptoms. It has 22 questions. The scale is composed of five points (0 to 4), with 0 being "not at all" and 4 being "a lot". On this scale, a total score of 24 or higher indicates "the need for clinical consideration for PTSD", a score of 33 to 36 is a "potential PTSD level", and a score of 37 or higher is clinically considered to indicate "to the extent that the immune system is not functioning properly". On this scale, a score higher than 17–18, is considered to indicate having partial PTSD symptoms, and if 24–25 or higher, it is considered to indicate having full PTSD symptoms [40]. The full PTSD assessment can be classified in detail as follows: a score of 25–39 is moderate PTSD, 40–59 is severe PTSD, and 60 or higher

is very severe PTSD [42]. The AIS comprises eight questions to evaluate the quantity and quality of sleep. This scale has four score points for each question (0 to 3), with 0 being "no problem at all" and 3 being "very serious". This scale is used to make a clinical judgment on the degree of sleep disorder using the sum of scores. If the total score is less than 4, it is deemed "maintaining a good sleep state." If it is between 4 and 5, it is "doubtful insomnia", and if it is 6 or more, it is considered that an "insomnia test is required".

The subject wore the heart rate sensor, which measured the peak-to-peak interval (r-r′ interval) during each intervention session with a Bluetooth-based communication application. The sensor was worn on the subject's chest according to the manufacturer's guidelines [43]. During psychotherapy, the application obtained the subject's heart rate information and calculated the HRV in real time. After psychotherapy, the HRV was calculated using the Open Source Python Toolbox for HRV [44,45]. Among the various components of HRV, the root mean square of successive RR interval differences (RMSSD), the standard deviation of the NN intervals (SDNN), the absolute power of the very-low-frequency band (VLF, 0.00–0.04 Hz), the absolute power of the low-frequency band (LF, 0.04–0.15 Hz), the absolute power of the high-frequency band (HF, 0.15–0.40 Hz), and the ratio of the LF-to-HF power (Ratio) were calculated and compared [44–46]. The calculation equations for the RMSSSD and SDNN are as follows [46]:

$$RMSSD = \sqrt{\frac{1}{N-1}\sum_{i=1}^{N}(RR_{i+1} - RR_i)^2} \qquad (1)$$

$$SDNN = \sqrt{\frac{1}{N-1}\sum_{i=1}^{N}(NN_i - \overline{NN})^2} \qquad (2)$$

where N is the number of measured values, RR is the time between two detected heartbeats, and NN is the time between two filtered heartbeats. The initial 3-min data for each session were discarded as artifacts. The HRV indexes neurocardiac function and is generated by heart–brain interactions and dynamic non-linear ANS processes [46]. The HRV is an emergent property of interdependent regulatory systems, which operate on different time scales to help us adapt to environmental and psychological challenges [46]. Higher values of RMSSD and SDNN signify higher top-down self-regulation (the ability to regulate behavioral, cognitive, and emotional processes) [47].

The change ratio of the HRV was assessed as well to check the subject's current state. The RMSSD and SDNN in this investigation were calculated using the measured heartbeat intervals for the latest five minutes. The RMSSD at minute 18 was calculated using the recorded heartbeat intervals from minute 13 to minute 18. If the subject started to self-report his traumatic event at minute 14, it could affect the HRV factors at minute 18 or minute 23. To calculate the change ratio of the HRV factors, the difference between the former five-minute HRV factors to the present five-minute HRV factors was divided by the present five-minute HRV factors. To check the effect of trauma telling at minute 14, the change ratio from the HRV factors at minute 13 to the HRV factors at minute 18 was calculated and noted as "Time 0". To track the slow effect of trauma telling at minute 14, the change ratio from the HRV factors at minute 18 to the HRV factors at minute 23 was computed and noted as "5 Min. later".

In summary, the "Time 0" change ratio was calculated as follows:

$$Time\ 0 = \frac{HRV_0 - HRV_{-5}}{HRV_0} \qquad (3)$$

and the "5 Min. later" change ratio was computed as follows:

$$5\ Min.\ later = \frac{HRV_5 - HRV_0}{HRV_5} \qquad (4)$$

where $HRV_{-5}$ is the HRV factor calculated using the heartbeat intervals from minute $-10$ to minute $-5$, $HRV_0$ is from $-5$ to 0, and $HRV_5$ is from 0 to 5.

### 2.4.2. Qualitative Analysis

For the qualitative analysis, a case analysis method was used [48]. This method enables understanding of the process by which a case develops in a unique and complex way in a specific situation or context by collecting in-depth data, including rich information sources appearing in the context. The psychotherapist encouraged the subject to express his emotions, thoughts, and attitudes, especially during visual observation of the subject's symptom changes in the intervention session. Each session was voice-recorded with the subject's consent, including the self-expression process. Dramatic emotion changes, self-reporting of traumatic events, and treatment with stabilization techniques were noted. The observation notes and transcript of the recorded session were analyzed. Our qualitative analysis was cross-checked three times by each of three experts officially qualified as, respectively, a mental health social worker supervisor, a disaster mental health specialist, and an expert belonging to the KCA. On the basis of these nine cross-checks, the authors' analysis and those of the three experts were found to show 90% correspondence.

## 3. Results and Discussion

### 3.1. Therapeutic Intervention and Symptom Change during the Proposed Program

To analyze the effectiveness of the therapeutic intervention on the symptoms of the subject in every session, the observation notes (dramatic changes in positive and negative emotions of the subject, self-reporting of his traumatic events, treatment with stabilization techniques) and the transcripts were analyzed qualitatively. The qualitative analysis was compared with the calculated HRV in each session. The comparison is outlined below.

### 3.1.1. Session One: Exploring Traumatic Events and CPTSD Symptoms, Creating a Safety Zone, and Training Stabilization

In the first session, traumatic events and CPTSD symptoms were first explored. After this, the stabilization treatment to establish a safety zone for relieving and self-managing the subject's symptoms was implemented. Figure 1 shows the 5 min RMSSD and the SDNN for the first session. At 15 min from the start of the first session, the subject self-reported his core traumatic event (the "T" remark). The abrupt negative emotional change (the "N" remark) and the abrupt drop in the RMSSD near the 15 min point appear to have been related to this self-reporting. However, the SDNN near the 15 min point increased in contrast to the RMSSD.

In the first session, the subject self-narrated his traumatic events at approximately minute 14 and minute 51 from the session start: "Hits for no reason...all my friends know." "I'm still scared when I run into kids who look like mutts on the way. When I think of their faces, I get angry...scared." The subject expressed negative emotions, such as shame and neglect, while recalling the above key traumatic events. The subject self-reported symptoms of CPTSD caused by these events, such as re-experience, avoidance, negative self-concept, and interpersonal difficulties. To address the core symptoms, four sets of safety-zone-making and stabilization techniques were performed from minute 35 to minute 52. When the subject recalled the traumatic event, he self-reported discomfort as "8" on the scale before the stabilization technique was implemented. After the stabilization technique was performed, he self-reported "4" for his discomfort level.

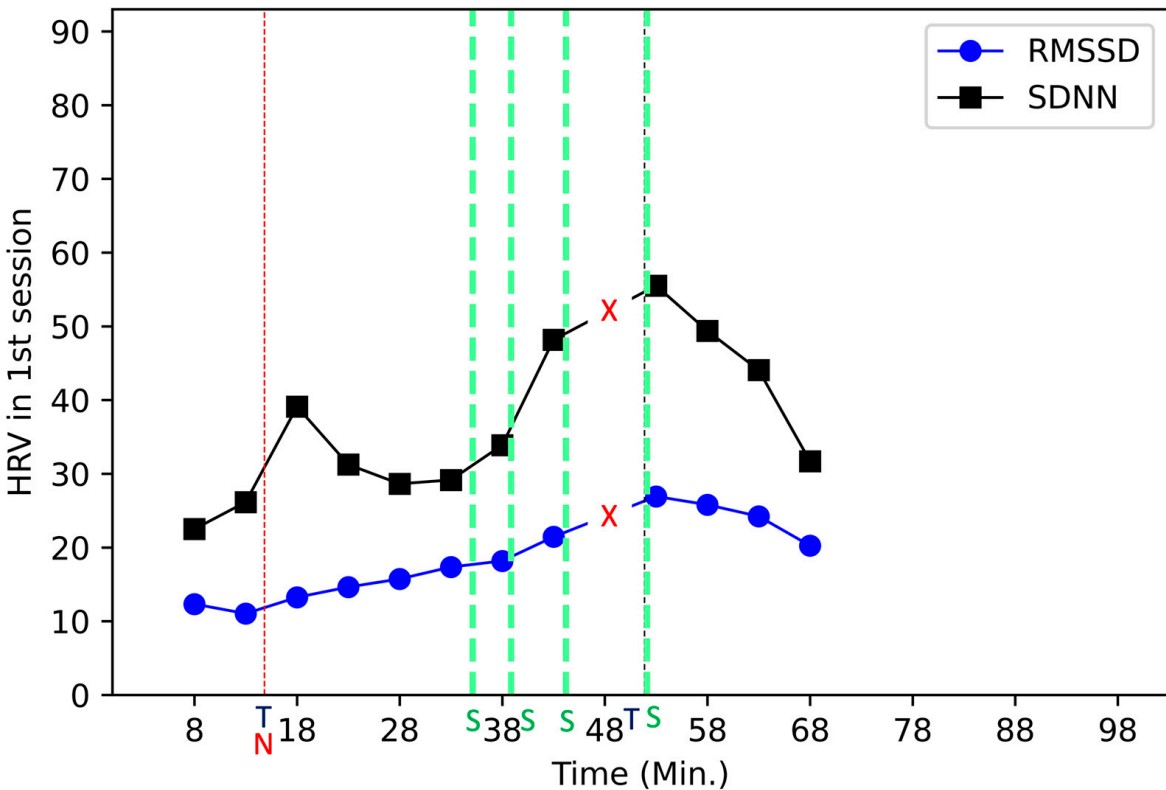

**Figure 1.** Every 5 min SDNN and RMSSD in the first session. The 3 min data from the session start were removed from being considered as the stabilization time. In the period from minute 43 to minute 48 from the session start, the sensor was so loose that the data were recorded as notably high. The "X" mark denotes that period. The "T" marks are the trauma self-narratings. The "N" mark is the negative emotion changing abruptly. The "S" marks are the stabilization techniques. In this session, the first trauma self-narrating and the negative emotion change happened at around minute 15 simultaneously. At around minute 52, the second trauma self-narrating and the last stabilization technique were performed together.

### 3.1.2. Session Two: Cognitive Reconstruction

The second session was the stage of restructuring the subject's distorted cognition by finding the core beliefs that had a significant impact on his core being. He identified his traumatic events with his other experiences. He self-narrated his traumatic events: "Sometimes I thought I might be mentally sick. When I see someone whom I don't like, I just want to kill them...so I thought I had a mental disease". "I'm just walking around...a random assault could happen to me". [What is the random assault?] "It's when suddenly, for no reason, someone just beats, assaults, and robs me." [Is that the same as your experience when a friend slapped the back of your head on the bus in high school?] "Isn't it the same thing?" [Can you tell me what you read in the newspaper about the random assault?] "It was an assault and threats with a knife".

This is a typical symptom of CPTSD victims: a negative self-concept and cognitive distortion cause the emotional changes. After the initial stabilization technique was implemented at the start of the session (approximately 5 min from the beginning), the subject self-reported his interpersonal experience until the middle of the second session. During this self-reporting period, the subject exhibited depression and an abrupt decrease in the SDNN (Figure 2).

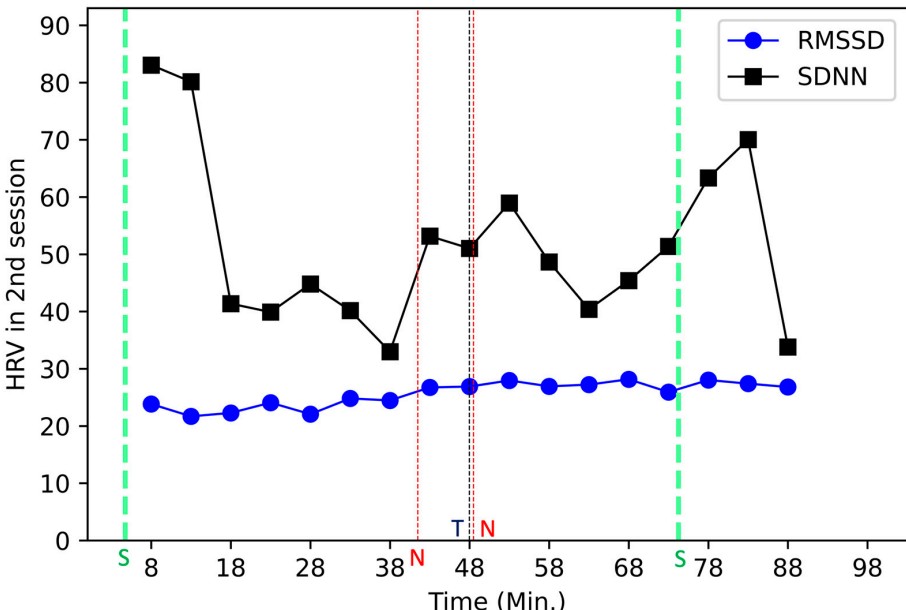

**Figure 2.** Each 5 min SDNN and RMSSD in the second session. The 3 min data from the session start are omitted as part of the stabilizing time. The "S" marks are the stabilization techniques. The "N" marks are the negative emotion changing abruptly. The "T" mark is the trauma self-narrating.

Immediately after the subject's self-narration of traumatic events at approximately 4 min, both the SDNN and RMSSD increased, as shown in Figure 3, which suggests that the self-narration induced the subject's psychological change. At approximately 74 min, the preferred sense of the subject among the five senses was explored. Based on the explored preferred sense, the second stabilization technique was performed. Psychological stability was self-reported as "2" in terms of the discomfort level, and increases in the SDNN and RMSSD were observed, as shown in Figure 2.

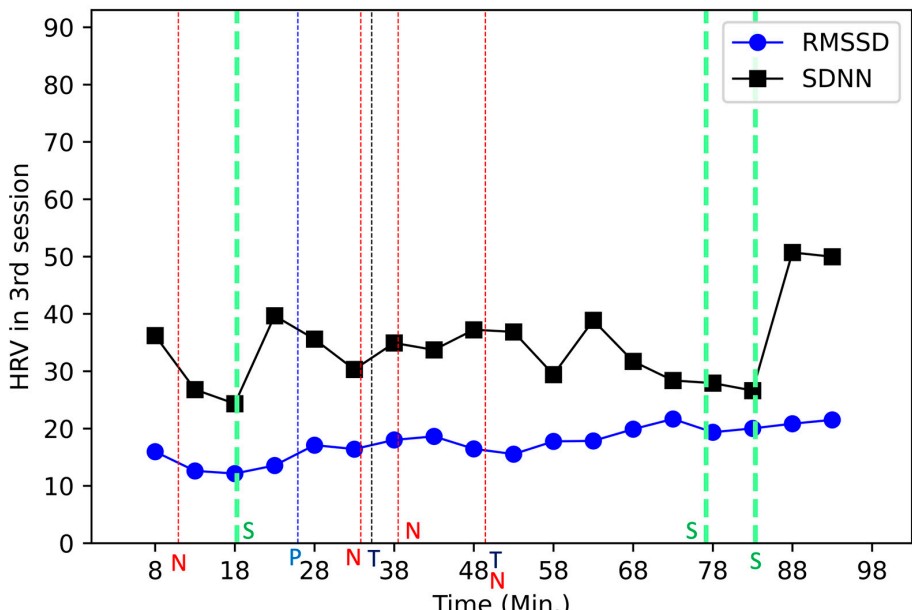

**Figure 3.** Each 5 min SDNN and RMSSD in the third session. The 3 min data from the session start were omitted as being part of the stabilizing time. The "N" marks are the negative emotions changing abruptly. The "P" mark is the positive emotion changing abruptly. The "S" marks are the stabilization techniques. The "T" marks are the trauma self-narratings. In this session, at around minute 49, the trauma self-narrating and the negative emotion change happened simultaneously.

The stabilization technique, a bottom-up process, produced an immediate response to the HRV changes compared to the top-down process, such as the cognitive intervention method. Since the cognitive changes of those who experience CPTSD are chronic symptoms, the process of changing distorted cognition and thinking is likely to be a more difficult process. Therefore, during the top-down process, a dramatic change in HRV was not observed, unlike in the bottom-up intervention.

3.1.3. Session Three: Sensory Reconstruction and Identifying Resources

In the third session, the subject clearly revealed his negative self-concept, which was not integrated within himself. At the start of this session, a rapid decline in both the SDNN and RMSSD was observed, as shown in Figure 3. At this point, the subject self-reported his unacceptable experience. In the transcript, this was described as follows: "Embarrassment is seven points. . . . Injustice is ten points. It's really terrifying. It's unfair I had to go through this experience". "Just think about it for a moment. . .I wouldn't have gone this far. . .I'm so sorry for him. . ..It's confusing. I'm sorry and I'm annoyed. . ... However, it was not that I was 100% kind, and he did that kind of thing afterward [after recalling the traumatic event]." "The maximum points, ten points (of discomfort). . ..Does this make sense? [a sigh and a deep breath. . .hesitating to talk]".

Since he had self-reported his negative self-experience at the beginning of the session, he self-narrated his traumatic symptoms up to the middle of the session. He scored his traumatic events as ten points, the maximum points. Although he reported negative events, as in Session 2, the previous session, he deconstructed and reconstructed his traumatic events in talking with the psychotherapist, which resulted in a symptom change. This process is a typical top-down one. In the process of discussing the traumatic events, the story is reassembled, which helps the CPTSD victim to have a new point of view toward the traumatic events. Here, it is suggested that the increasing trend of both the SDNN and RMSSD from approximately minute 18 to minute 50, as shown in Figure 3, is related to these verbal reports.

In this session, the stabilization technique, which involved exploring resources, was performed. First, the subject was to hug himself, identify an external resource (someone beside himself), and then experience acceptance by the external resource. Next, the stabilization technique based on the subject's preferred sense was performed, in a similar way to the second session. The subject rated his experience, which ranged from ten points to six points after the second stabilization and to three points after the third stabilization.

3.1.4. Session Four: Confirming One's Own Change, Exploring Self-Management Plan, and Projecting One's Own Future (Going into the World)

In the last session, the subject confirmed his own changes and explored self-management methods. These results appear to be related to the abrupt positive emotion changes at approximately minute 5 and minute 42, as shown in Figure 4. The HRV at both moments exhibited increasing tendencies, except for the SDNN at 5 min. The subject described his positive emotion accordingly: [What kind of person do you think you are?] "I think I am growing a bit. . ..I could say it is unnecessary to be upset. Let's talk just about this situation".

The stabilization technique in the last session utilized a timeline based on the subject's preferred sense. As the intervention program progressed, the subject integrated his separate sense of self. He practiced maintaining these changes in the future. He talked about a problem centered on himself as the victim at approximately minute 45 (Figure 4). This re-experience is a phenomenon that can happen to those who have experienced CPTSD. Therefore, through a future projection technique, the subject's coping methods were checked for a future situation. After the projection process, breathing training and imagery training work were conducted. During these sessions, the subject experienced positive sensations of the five senses through light stream mediation. Through a body

scanning technique, he verbally reported a "warm and cozy sensation". The HRV, as shown in Figure 4, increased as well.

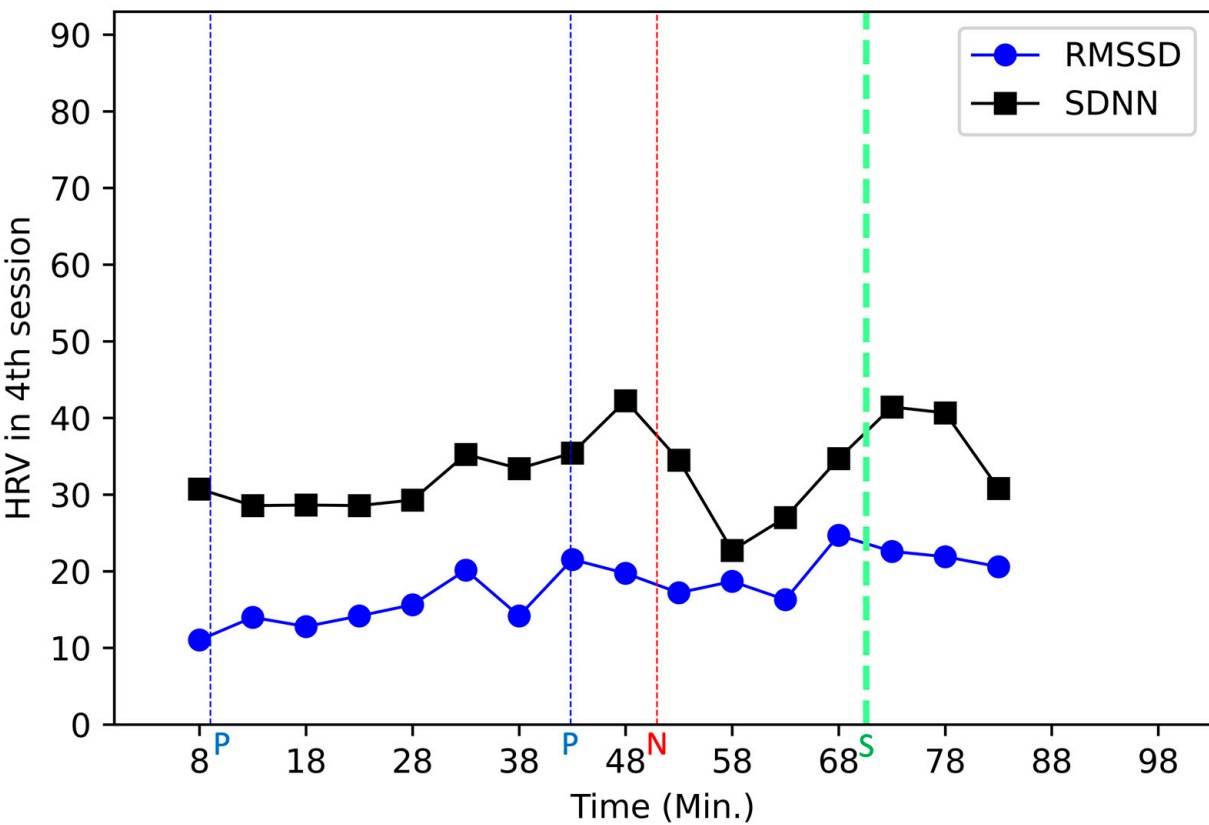

**Figure 4.** Each 5 min SDNN and RMSSD in the fourth session. The 3 min data from the session start are omitted as being part of the stabilization period. The "P" marks are the positive emotion changing abruptly. The "N" mark is the negative emotion changing abruptly. The "S" mark is the stabilization technique.

During the future projection process, the subject assessed himself accordingly: "Compared to five years ago, I am doing better.... First of all, I'm getting older and getting to know the world more. I grew up". [Are you punishing yourself like you did before?] "I don't think so". [What do you want to say to yourself in five years?] "Something cool. My future is wonderful. I would like to inherit something valuable".

### 3.2. Correlation between the Intervention Event and the HRV Change Ratio

During all four sessions, the subject self-reported his traumatic event five times. Among the five trauma tellings, abrupt negative emotion changes within five minutes of the trauma tellings were observed four times. In the case of the only trauma telling not related to abrupt negative emotion change, the psychotherapist performed the stabilizing technique right after the trauma telling. Among the ten stabilizing techniques used in all four sessions, abrupt positive emotion change within five minutes was observed once. The trauma telling appears to have altered the subject's psychological status.

To statistically evaluate the relationship between the intervention events (trauma telling, stabilization technique, positive emotion change, and negative emotion change) and the subject's psychological and physical status, the change ratios of the calculated HRV factors (RMSSD, SDNN, VLF, LF, HF, and Ratio) were analyzed. The change ratios of the HRV factors were used instead of their absolute values because the change trends are more meaningful than the absolute values.

First, with respect to the cases marked as no intervention event, the change ratios of the HRV factors were calculated. In all four sessions, the time-zero HRV change ratios of

the non-noteworthy events were counted as 39. The five-min-later HRV change ratios were 23. The standard deviations of the RMSSD and SDNN were under 0.327, much smaller than the other standard deviations. The standard deviation range of the other HRV factors ranged from 0.489 to 1.803 and their average was 0.934. The average change ratios of the RMSSD and SDNN were $-0.3\%$ and $-10.2\%$ at time 0 and 1.1% and $-11.9\%$ five minutes later, respectively. The RMSSD showed almost no change during the ordinary status, but the SDNN showed a gradual decrease of approximately 10%.

Figure 5 exhibits the change ratio of the HRV factors when the subject described his traumatic event at time 0. In all four sessions, the trauma telling happened five times. The RMSSD and SDNN showed only reliable standard deviations. The SDNN increased by 10% at time 0 of the trauma telling and then decreased by 10% five minutes later. In relation to the average SDNN change ratio of the ordinary status (around $-10\%$), it appears that parasympathetic change occurred due to the trauma telling and then returned to the ordinary status. Considering the above-mentioned relation between the trauma telling and the abrupt negative emotional change, the SDNN change ratio according to the trauma telling can be explained. Even though the standard deviation of the RMSSD was the least among all the HRV factors, its change ratio (under 10%) was negligible compared to its change ratio at the ordinary status (around 0%).

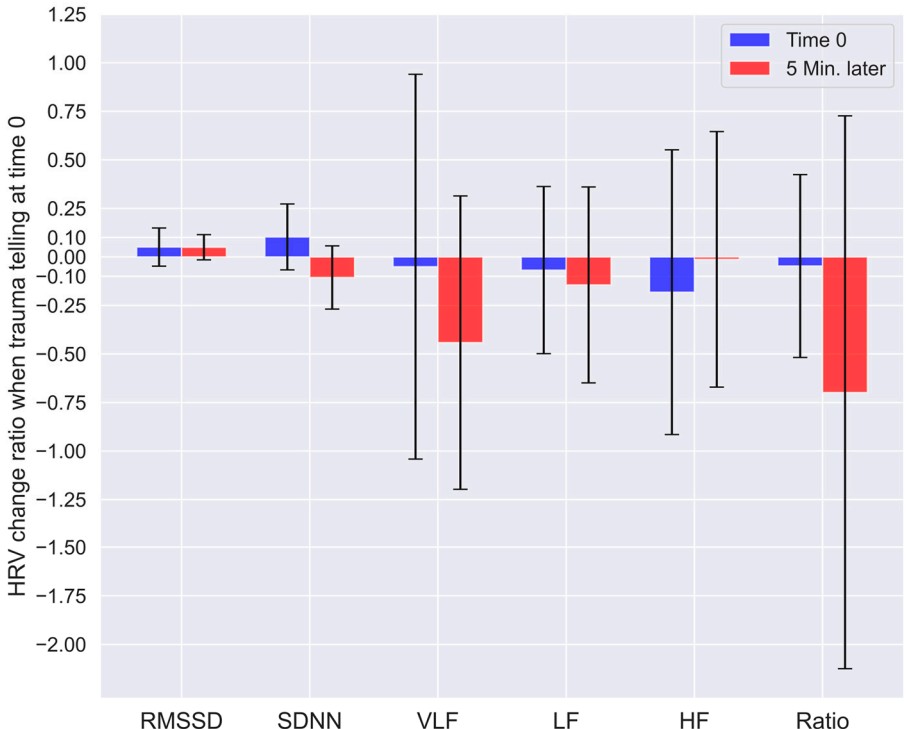

**Figure 5.** Change ratio of the HRV when the subject described his trauma at time 0. During all the 4 sessions, the subject described his trauma 5 times.

The change ratio of the HRV factors when the psychotherapist performed the stabilizing technique at time 0 was as shown in Figure 6. In all four sessions, the stabilizing technique was performed ten times. The RMSSD and SDNN showed only reliable standard deviations. The SDNN increased by about 23% at time 0 and then exhibited almost no change five minutes later. It is inferred that the stabilizing technique facilitated autonomic flexibility. Even though the standard deviation of the RMSSD was the least among all the HRV factors, its change ratio (under 10%) was negligible compared to its change ratio at the ordinary status (around 0%).

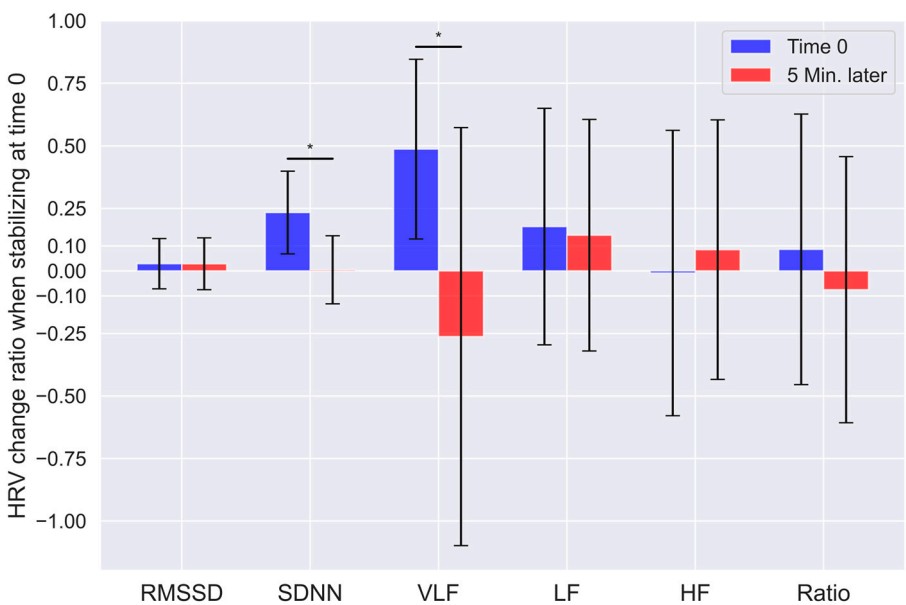

**Figure 6.** Change ratio of the HRV when the stabilizing technique was performed at time 0. Over the 4 sessions, the stabilizing technique was used 10 times. The "*" mark indicates that the *p* value is under 0.05.

Figure 7 shows the change ratio of the HRV factors when the subject's abrupt positive emotion change was observed at time 0. In all four sessions, a positive emotion change was observed four times. The RMSSD and SDNN showed only reliable standard deviations. The RMSSD at time 0 only increased by about 25%. The SDNN at time 0 decreased by about 4.4%. For positive emotional change in all the sessions, the RMSSD changed by over 10%. Considering this evidence, the 25% RMSSD increase at time 0 does not appear to consistently represent the subject's status at that time.

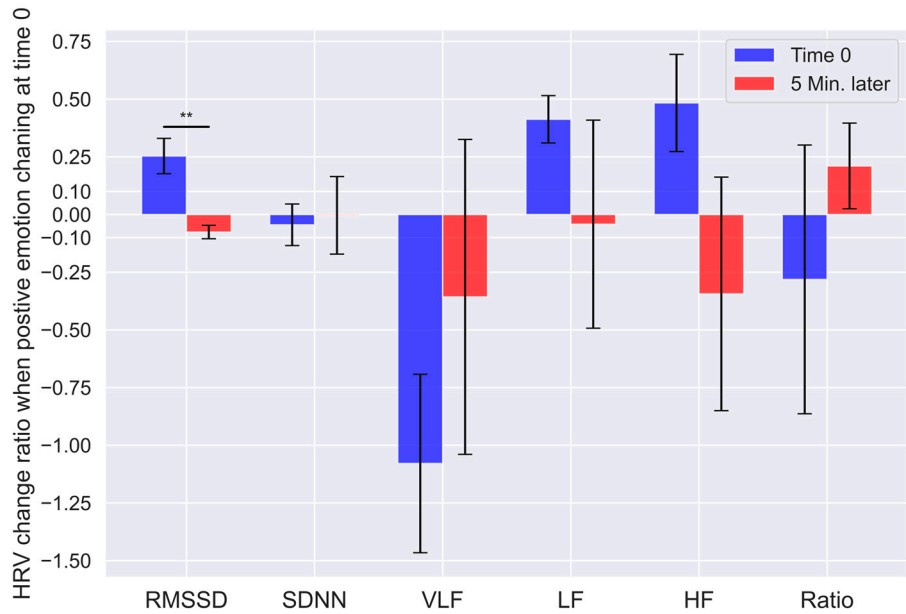

**Figure 7.** Change ratio of the HRV when the subject's positive emotion change was observed at time 0. Over the 4 sessions, positive emotion change was observed 3 times. The "**" mark indicates that the *p* value is between 0.001 and 0.05.

The change ratio of the HRV factors when the subject's abrupt negative emotion change was observed at time 0 are shown in Figure 8. In all four sessions, a negative

emotion change was observed eight times. The RMSSD and SDNN only showed reliable standard deviations. The SDNN increased by about 4.5% at time 0 and then decreased by approximately 17% five minutes later because of the negative emotion change. Considering the average SDNN change ratio of the ordinary status (around −10%), the 4.5% increase was a significant change, and the 17% SDNN decrease appeared to be closer to the ordinary status. It is inferred that the abrupt negative emotion change caused the SDNN to increase at time 0. Even though the standard deviation of the RMSSD was the least among all the HRV factors, its change ratio (under 10%) was negligible compared with its change ratio at the ordinary status (around 0%).

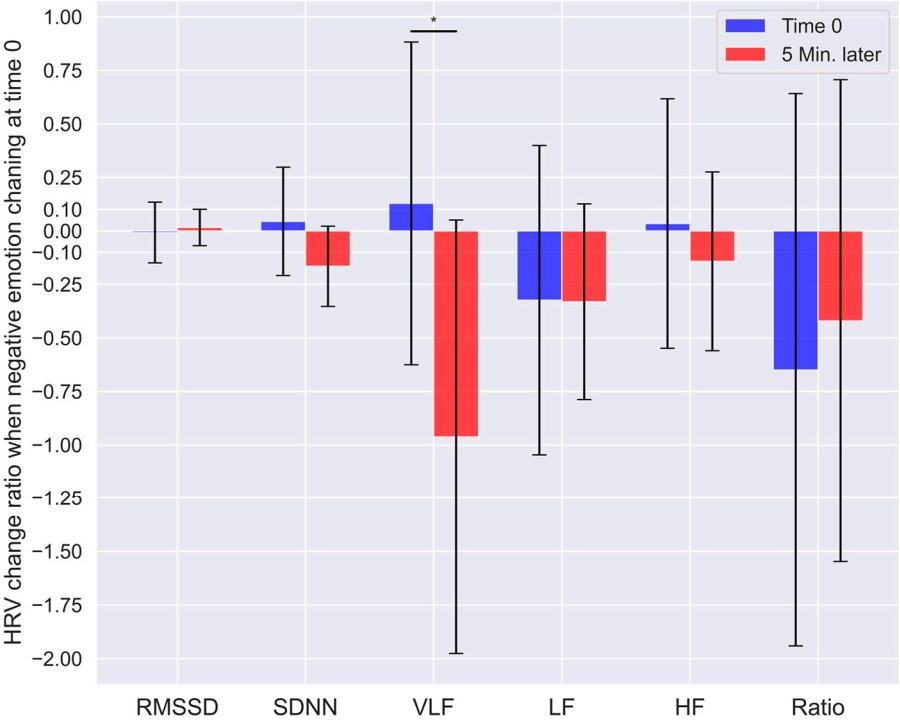

**Figure 8.** Change ratio of the HRV when the subject's negative emotion change was observed at time 0. Over the 4 sessions, negative emotion change was observed 8 times. The "*" mark indicates that the *p* value is under 0.05.

### 3.3. Change in the IES-R-K and AIS Scale before, after, and 10 Months after the Proposed Program

Table 1 shows the scale changes in the IES-R-K and AIS before, after, and 10 months after the proposed program. The IES-R-K values were 60 points in the pre-survey, 43 points in the post-survey, and 40 points in the follow-up survey, indicating that the level was lowered from a very serious level before the program to a serious level after the program. The IES-R-K changed to 43 points in the post-survey, which represented a low level in the severe range (40-59 points), and this status was maintained until the follow-up survey (40 points). With regard to the differences by subfactors of the IES-R-K, re-experiencing decreased the most from 23 to 15 points, followed by avoidance from 24 to 17 points, and sleep and dissociation from 14 to 10 points, with hyperarousal decreasing from 21 to 20 points. The AIS change indicated that the quality of sleep improved from 10 points in the pre-survey, to 7 points in the post-survey, and finally to 6 points in the follow-up survey.

**Table 1.** The scale change in the IES-R-K and AIS before, after, and 10 months after the proposed program (a lower score is more positive).

| | | Before | After | 10 Months After |
|---|---|---|---|---|
| IES-R-K (Full score: 88) | Re-experiencing | 23 | 10 | 15 |
| | Hyperarousal | 21 | 19 | 20 |
| | Sleep, emotion, dissociation | 14 | 15 | 10 |
| | Avoidance | 24 | 17 | 17 |
| | Summation | 60 | 43 | 40 |
| AIS (Full score: 24) | | 10 | 7 | 6 |

## 4. Discussion

During the intervention program, the subject's RMSSD and SDNN were monitored every five minutes (Figures 1–4). The RMSSD and SDNN showed a similar trend in sessions 1, 3 and 4. Only at the second session did the SDNN fluctuate more dramatically compared to the RMSSD. Despite the similar trends in sessions 1, 3 and 4, the change variation of the SDNN in the identical session was much bigger than that of the RMSSD. With regard to monitoring the subject's state, the more variable factor, the SDNN, is more appropriate.

The first 3 min of heart rate data in every session were not included as they were considered to reflect the stabilization of the subject. Despite of this stabilizing time, the coefficients of variation (CV) of the first RMSSD and SDNN (calculated on minute 8) in every session were 34.2% and 68.9%. The CVs of the RMSSD and SDNN right after the subject's trauma telling were 31.9% and 21.1%. This implies that it is inappropriate to compare the values with each other because of the fluctuation associated with the same event. Various researchers have recommended careful comparison of short-term HRV readings [46].

Therefore, we compared the change ratios of the HRVs. The comparison revealed that trauma telling had a significant impact on the subject's psychological status, as evidenced by the observed abrupt negative emotion changes within five minutes of the trauma tellings and the 10% SDNN increase at the time of the trauma tellings (Figures 5–8). This suggests that reliving the traumatic event through self-reporting can affect the subject's ANS as well as inducing emotional distress. The abrupt negative emotion changes followed by the trauma tellings increased the SDNN by about 10.5%, which was about twice the average SDNN increase (4.5%) at the time of all the abrupt emotion changes. The abrupt emotion changes followed by the trauma telling appeared to regulate the subject's ANS to a greater extent.

The occurrence of abrupt negative emotion changes had a notable impact on the SDNN, which increased by approximately 4.5% on average at the time of the negative emotion change (Figures 5–8). However, this increase was followed by a decrease of approximately 17% five minutes later, indicating a return to the subject's ordinary state. Furthermore, the stabilizing technique performed by the psychotherapist showed promising results in promoting autonomic flexibility. The change ratios of the HRV factors demonstrated an increase in SDNN by approximately 23% at the time of intervention, indicating a positive effect on the subject's physiological response. This suggests that the stabilizing technique may have helped the subject regulate their autonomic nervous system and to achieve a more balanced state.

Additionally, the occurrence of abrupt positive emotion changes was observed to influence the HRV factors in different ways (Figures 5–8). While the RMSSD showed a considerable increase of around 25% at the time of positive emotion change, the other HRV factors exhibited decreases of under 10%. This indicates that the HRV factors may not be

highly sensitive to transient positive emotional fluctuations. Therefore, relying solely on the RMSSD as a real-time indicator during psychological counseling may not be reliable.

Overall, the findings highlight the importance of considering different interventional events and their corresponding effects on a subject's psychological and physiological status. While trauma telling and stabilizing techniques showed significant associations with emotional and autonomic responses, the impact of positive emotion changes was less pronounced, and the use of RMSSD as a real-time indicator may be limited. These findings contribute to our understanding of the complex dynamics between interventional events and a subject's well-being, emphasizing the need for comprehensive assessments in psychological counseling settings.

Despite these positive findings, this research has some limitations: Only one subject participated in this study; recruiting more subjects would be required to statistically evaluate the usefulness of real-time HRV monitoring in psychotherapy. Along with the subject's ANS, their respiration could also affect the HRV values [49]. In order to remove the effect of respiratory sinus arrhythmia at the measured HRV, independent respiration measurement is required simultaneously with HRV measurement.

## 5. Conclusions

In this paper, a practical intervention program was proposed to help a subject to lead a normal daily life through cognitively restructuring their CPTSD symptoms by applying cognitive and physical interventions. The findings included the following: First, using real-time HRV, the psychotherapist could easily check the subject's current state. Second, the HRV showed a significant relationship with the intervention techniques, especially the change ratios of the HRV. Therefore, HRV can be utilized to check a subject's current state and assist in the choice of personalized intervention techniques among psychotherapy methods, such as the bottom-up method and the top-down method. This approach may overcome the limitations of subject self-reporting. This real-time HRV-assisted psychotherapy approach can be utilized much more frequently in online therapy, especially during pandemic periods. Currently, we are planning a more sophisticated investigation for a CPTSD patient cohort.

**Author Contributions:** Conceptualization, B.I. and K.-i.K.; methodology, B.I., J.K., T.K. and K.-i.K.; software, B.I., J.K. and T.K.; validation, K.-i.L. and K.-i.K.; writing—original draft preparation, B.I. and K.-i.K.; writing—review and editing, K.-i.L. and K.-i.K.; visualization, K.-i.K.; project administration, K.-i.L. and K.-i.K.; funding acquisition, K.-i.K. All authors have read and agreed to the published version of the manuscript.

**Funding:** This research received no external funding.

**Institutional Review Board Statement:** The study was conducted in accordance with the Declaration of Helsinki and was approved by the Institutional Review Board of University of Ulsan, Ulsan, Republic of Korea (IRB approval number: 1040968-A-2020-016).

**Informed Consent Statement:** Informed consent was obtained from all subjects involved in the study.

**Data Availability Statement:** The data presented in this study are available in article.

**Conflicts of Interest:** Author Bohye Im was employed by the company Daegu Maum-in Counseling Centre. The remaining authors declare that the research was conducted in the absence of any commercial or financial relationships that could be construed as a potential conflict of interest.

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
