# Peer review of "Utilizing Real-Time Heart Rate Variability during Psychological Intervention Program for Complex Post-Traumatic Stress Disorder: A Case Studyâ€"

_applsci, doi:10.3390/app14010004_

Round 1
Reviewer 1 Report
Comments and Suggestions for Authors
Although I failed to find the Authors rebuttal, Authors have added new parts, in particular an excerpt at p. 3 of the current version of the manuscript (lines 107-118).
A second excerpt was addedd at p. 4 of the current version of the manuscript replaying my raised question about the design of the presented study (lines 155-162).
They had added a brief explication of the absence of diagnostic critera In Korean societies to discriminate CPTSD symptoms from PTSD (p. 4, lines 180-181).
More information of the statistical analysis was addedd at p. 10 about the change ratios of the HRV factors calculated using the measured heart beat intervals for the latest five minutes. Furthermore, all the analysis of the relation between the interventional event and the HRV change ratio was modified along the stastistical analysis paragraph (at 10-13 pp).
Tables 1 was appropriately revised.
Globally, I think that the quality of the paper was improved with the revision.
Author Response
Dear Reviewer 1,
Thank you so much for your comments about the improvement in our manuscript.
Simultaneously, authors are sorry that our rebuttal letter was missed.
We don't know why it happened. We expect it might be a technical problem.
If you want, we will send it.
Thank you again.
Sincerely,
On be half of authors,
Kyo-in Koo,
Corresponding author.
kikoo@ulsan.ac.kr
Reviewer 2 Report
Comments and Suggestions for Authors
The study explores the use of HRV parameters during an intervention for CPTSD in one participant. Despite the importance of integrating physiological/biological measures in clinical practice, I found some issues in this study. The main limitation is that it was a case study. The presented results are promising, but I prefer to test the study hypothesis on at least 15/20 subjects (as a pilot study). Both the diagnosis of CPTSD and HRV have considerable variability in each definition (i.e., CPTSD patients did not have all the same symptoms, and HRV is a parameter affected by many covariates) to report the results only on one person. The second main issue was related to the collection and analysis of HRV data. It was not reported if the HRV portable device was validated to measure these parameters and have the same reliability as an ECG assessment. Moreover, all the analyses of the cardiac signal do not account for – at least – variability related to age, BMI, or smoke status (if applicable). Using one subject makes it impossible to do, but reporting the results for only one person could potentially lead to erroneous conclusions. Again, related to HRV, it is unclear to me the pre-processing of the RR intervals and how artefacts are managed (a part to exclude the first three minutes of the recording). Since other studies reported different interventions for PTSD and integrated the HRV measurement on a bigger sample size, I have few doubts about how this study could add to the literature. If the authors increase the sample size and make other analyses accounting for the covariance of some variables, I think it could be an excellent work.
Reviewer 3 Report
Comments and Suggestions for Authors
Although I consider that the (single) study is well designed, structured and organized, there is no doubt that the preliminary results presented here need to be appropriately supported not only from a statistical point of view but also from a scientific point of view. Validation of results using a statistically reliable sample size is required, this will not only allow the consolidation of results but also the possibility of establishing comparisons with studies carried out in the same scope. In fact, this is a weakness of the results presented here, they are not compared with the results obtained by other authors. However, the authors identify in section 4 some of the limitations of the work carried out, which demonstrates that they are aware of the weaknesses already mentioned. As such, and as it is my opinion that the work is well structured and presented clearly, it deserves to be published and perhaps with this possibility it will gain sufficient notoriety to allow it to access samples with a number of individuals with statistical representativeness, and in the future, to allow the establishment of a procedure protocol.
Author Response
Dear Review 3,
Authors appreciate your recommendation to publish our manuscript.
Thank you again.
Sincerely,
On be half of authors,
Kyo-in Koo,
Corresponding author.
kikoo@ulsan.ac.kr
Reviewer 4 Report
Comments and Suggestions for Authors
I read with interest the manuscript entitled "Utilizing real-time heart rate variability during psychological intervention program for complex post-traumatic stress disorder: A case study"
First of all, I suggest that you submit the manuscript in the case report category, not the article category.
Correct keywords according to MeSH terms of interest.
Although the introduction contains all the relevant information, I suggest that you write it more succinctly and concisely, and at the end of the same, briefly and clearly state the goal and hypothesis of the research. Do not state the research methodology in the introduction.
Case reports should include relevant positive and negative findings from history, examination and investigation. Case reports should include an up-to-date review of all previous cases in the field. Authors should follow the CARE guidelines and the CARE checklist should be provided as an additional file (https://www.care-statement.org/checklist).
It is unnecessary to state information such as ethical approval within the text and at the end of the manuscript. It is enough only at the end.
Add the missing parts according to the CARE guidelines. The discussion is insufficient, so please write it again in its entirety.
Comments on the Quality of English LanguageModerate editing of English language required.
Round 2
Reviewer 2 Report
Comments and Suggestions for Authors
I would like to thank the author for their responses and adjustments to the paper. It could be a reasonably valuable work if published as a “report case”. Thank you.
Author Response
Dear Reviewer 2,
Authors appreciate your valuable recommendation. We changed our article type as "Case Report".
Sincerely,
Kyo-in Koo
Reviewer 4 Report
Comments and Suggestions for Authors
I don't see that the template's manuscript category has changed. Also, the case presentation should be in the form: introduction, case presentation, discussion, and conclusion. Please implement the above suggestions so that the manuscript category matches the case presentation.
Comments on the Quality of English LanguageModerate editing of English language required.
Author Response
Dear Reviewer,
Authors appreciate your valuable recommendation. Following your comment, we changed the section name as "2. Case Description and Methods". Please find in the revised manuscript.
Round 3
Reviewer 4 Report
Comments and Suggestions for Authors
Unfortunately, you did not apply the well-intentioned suggestions.
Comments on the Quality of English LanguageModerate editing of English language required.
Author Response
Authors appreciate your constructive recommendation. Following your comment, we changed structure of the manuscript thoroughly. Please find the changes in the revised manuscript.
The special editing agency checked our manuscript. We attached the certificate as well.
We believed we revised our manuscript sincerely following your comments. However, if something required more, could you describe more specifically?
